# Numerical Investigation on the Influence of Water Vapor Ionization on the Dynamic and Energy Deposition of Femtosecond Ultraviolet Laser Filamentation in Air

**Qingwei Zeng**, **Lei Liu** \*, **Kejin Zhang, Shuai Hu, Taichang Gao, Chensi Weng** and **Ming Chen**

College of Meteorology and Oceanography, National University of Defense Technology, Nanjing 211101, China; zengqingwei519@yahoo.com (Q.Z.); zkj3047@163.com (K.Z.); hushuai2012@hotmail.com (S.H.); gaotc2009@yeah.net (T.G.); WCS19951114@163.com (C.W.); ChenMing1992@hotmail.com (M.C.)

\* Correspondence: liuleidll@gmail.com; Tel.: +86-25-8083-0652

**Abstract:** The effects of water vapor ionization on the nonlinear propagation of femtosecond laser pulses with a 248 nm wavelength are numerically investigated in this paper. It is found that ionization of $H_2O$ molecules plays a significant role in air ionization, which seriously affects the dynamic and energy deposition of filamentation. The propagation of femtosecond pulses in air with different humidity levels are compared. The total number of electrons and total deposited pulse energy increase with the humidity increases. However, they tend to be saturated in high humidity conditions. Results presented here are conducive to characterizing the long-range propagation of filaments under atmospheric conditions.

**Keywords:** ultrafast nonlinear optics; atmospheric propagation; self-focusing; multiphoton absorption; water vapor

## 1. Introduction

In the past decade, investigation of the formation of light filaments in air has become a topic of active experimental and theoretical research due to their various potential atmospheric applications, such as remote sensing of chemical/biological agents [1] and meteorological elements [2], high-voltage discharge guiding [3], air waveguiding [4], and weather-related applications [5–7].

The widely acceptable physical mechanism that sustains the formation of light filaments is the result of a dynamic balance between the Kerr self-focusing effect and the plasma defocusing effect [8]. Firstly, the optical Kerr effect focuses the pulse and then the beam intensity experienced obviously increases. When the beam intensity becomes sufficiently large, it ionizes the medium and creates an electron plasma [9]. Then, the electron plasma defocuses the trail of the pulse and strongly shortens its leading edge around and beyond the nonlinear focus [10].

Many filamentation-based applications rely on long distance propagation in atmosphere [11]. It is well known that real atmospheric air is composed of various types of gas molecules, which vary in content. In previous theoretical simulation studies, the ionization of $N_2$ and $O_2$ was usually only considered since they are the main components of air gas [12–14]. The ionization of water vapor molecules is often neglected because their concentration is usually too low. However, influences of water vapor on multiple physical effects during the filamentation process are expected to be very different when a laser pulse wavelength increases [15]. Up to now, only a few studies have considered the influence of relative humidity on the propagation of femtosecond pulses. Manuilovich et al. [16]

showed that the spatiotemporal shape of femtosecond pulse at a central wavelength of 800 nm broadens and deforms under high relative humidity conditions. Simulation studies also showed that physical effects like dispersion and linear absorption become more and more important in the formation and propagation dynamics of 10 μm filaments with the increase of relative humidity [15]. More recently, Shutov et al. [17] experimentally identified that the photoionization of water vapor acts as the dominant ionization process in atmospheric air for femtosecond laser pulses at a 248-nm central wavelength and the relevant ionization coefficients were also measured. With increasing air humidity, the plasma channels generated by ultraviolet laser pulses in air become longer and wider [18]. However, the detailed dynamic evolution of UV filamentation and its energy deposition characteristics in an atmospheric environment with different humidity levels has rarely been reported. Allowing for the reported valuable applications of UV laser filaments in meteorology [19–21], it is also necessary to evaluate how water vapor will affect the transmission of femtosecond lasers in a cloud environment.

In this paper, we numerically investigate the influences of ionization of water vapor on nonlinear propagation of ultraviolet (UV) filaments. The formation and propagation dynamics of UV filaments under different humidity conditions are studied in detail. The remainder of this paper is organized as follows: the model used to describe the propagation of femtosecond pulse in gas medium is introduced in Section 2; in Section 3, the evolution of filament along propagation distance z in different gas media is investigated; and the conclusions are summarized in Section 4.

## 2. Methods

To investigate the propagation dynamics of a femtosecond pulse in a gas medium, we modeled the propagation of a laser pulse with the nonlinear Schrödinger equation (NLSE), which governs the evolution of the electric field envelope $E(x,y,z,t)$ of the pulse in the reference frame moving at the group velocity $v_g = \partial\omega/\partial k\big|_{\omega_0}$ [9]:

$$\frac{\partial E}{\partial z} = \frac{i}{2k_0}\Delta_\perp E - \frac{ik''}{2}\frac{\partial^2 E}{\partial \tau^2} + i\frac{\omega_0}{c}n_2|E|^2 E - k_0\frac{\omega_{pe}^2}{\omega_0^2}E - \sum_{s=\mathrm{N_2,O_2,H_2O}}\frac{\beta_s^{(K_s)}}{2}|E|^{2K_s-2}E \tag{1}$$

where $E(x,y,z,t)$ represents the envelope of the electric field along propagation direction z, and $\tau$ refers to the retarded time variable, $t-z/v_g$, according to group velocity as the reference system. $k_0 = 2\pi/\lambda_0$ and $\omega_0 = 2\pi c/\lambda_0$ are the central wave number and the angular frequency of the carrier wave corresponding the wavelength $\lambda_0 = 248$ nm, respectively. The Laplacian operator $\Delta_\perp^2 = \frac{\partial^2}{\partial x^2} + \frac{\partial^2}{\partial y^2}$ denotes the beam transverse diffraction. The second terms on the right-hand side of Equation (1) account for group velocity dispersion (GVD) with a coefficient of $k'' = 1.21 \times 10^{-28}$ ($\mathrm{s^2/m}$). The Kerr effect is related to $n_2 = 8.0 \times 10^{-19}$ $\mathrm{cm^2/W}$ [12], which denotes the nonlinear refraction index of air. Plasma defocusing is described by the plasma oscillation frequency, $\omega_{pe} = \sqrt{q_e^2\rho/m_e\varepsilon_0}$ ($q_e$ is the electron charge, $m_e$ is the electron mass, and $\rho$ is the free electron density). The last term in Equation (1) accounts for multiphoton absorption (MPA). The MPA term accounts for energy absorption due to multiphoton ionization [9]. The coefficient $\beta_s^{(K_s)} = K_s\hbar\omega_0\rho_{at,s}\sigma^{K_s}$ is related to the multiphoton ionization coefficient, which corresponds to the number of K-photons needed to ionize the corresponding gas molecule.

The evolution equation of the electron density $\rho_{es}$ is governed by:

$$\frac{\partial \rho}{\partial t} = \sum_s \left( \frac{\beta_s^{(K_s)}}{K_s\hbar\omega_0}|E|^{2K_s} + \frac{\sigma}{U_s}\rho_{es}|E|^2 \right)\left(1 - \frac{\rho_{es}}{\rho_{at,s}}\right) \tag{2}$$

where $\sigma = 5.1 \times 10^{-22}$ is the cross section for inverse Bremsstrahlung, $\hbar = h/2\pi$ is Planck constant, $U_s$ is ionization potential of the gas, and $\rho_{at,s}$ denotes the initial neutral gas atom density. The coefficients used in Equations (1) and (2) are listed in Table 1.

**Table 1.** Parameters used in the model. We adopted most of the theoretically calculated parameters for $O_2$ and $N_2$ from Reference [9]. The effective multiphoton ionization cross section $\sigma^K$ for $H_2O$ molecules is taken from Reference [17].

| Gas Types | Symbol (Units) | Values [9] | The Measured Values |
|---|---|---|---|
| $O_2$ | K $\sigma^K$ ($s^{-1}$ $m^{2K}/W^K$) $U_i$ (eV) | 3 $1.35 \times 10^{-40}$ 12.07 | 3 $1.5 \times 10^{-40}$ 12.07 |
| $N_2$ | K $\sigma^K$ ($s^{-1}$ $m^{2K}/W^K$) $U_i$ (eV) | 4 $3.22 \times 10^{-60}$ 15.58 | 3 $2.4 \times 10^{-41}$ 15.58 |
| $H_2O$ | K $\sigma^K$ ($s^{-1}$ $m^{2K}/W^K$) $U_i$ (eV) | - - - | 3 $6.8 \times 10^{-38}$ 12.62 |

The initial pulse investigated in this paper is modeled by a Gaussian beam:

$$E(x, y, z = 0, t) = \sqrt{\frac{2P_{in}}{\pi w_0^2}} \exp\left(-\frac{x^2 + y^2}{w_0^2} - \frac{t^2}{t_p^2}\right) \tag{3}$$

where $P_{in}$, $w_0$, and $\tau_p$ are the input peak power, initial pulse radius, and the pulse duration, respectively.

A standard split step Crank-Nicholson scheme is used to integrate the linear part of Equation (1) along the propagation axis z and the fourth-order Runge–Kutta method is employed to solve the evolution equation (Equation (2)) [14].

## 3. Results and Discussion

We first considered incident laser pulses with $w_0$ = 1 mm, $\tau_p$ = 90 fs, and $P_{in}$ = 10$P_{cr}$. The total atmospheric air molecule density $\rho_{at}$ was assumed to be $2.7 \times 10^{25}$/$m^3$. For simplicity, dry air was assumed to only consist of $N_2$ and $O_2$ at a ratio of 0.78 to 0.21. For wet air, we imposed a composition of 78.0% $N_2$, 21.0% $O_2$, and 1.0% $H_2O$ (gas).

Figure 1 presents the on-axis intensity and on-axis electron density along the propagation axis. The solid blue and green lines represent the results for pulses in dry and wet air conditions, respectively. The multiphoton ionization cross section $\sigma^K$ for $N_2$, $O_2$, and $H_2O$ are taken from the measured results by Shutov [17], as listed in Table 1. The black dash line represents the simulated results for dry air when the cross section $\sigma^K$ for $N_2$ and $O_2$ were obtained from the theoretical formula by Couairon and Mysyrowicz [9].

As shown in Figure 1a, the beam intensity in both dry air and wet air first increased rapidly at about z = 2.0 m, which implies that the water vapor had no influence on the self-focusing position (defined as the distance between the initial transmission position of the beam and the position where the on-axis light intensity increases sharply). Almost simultaneously, the on-axis electron density also obviously increases, as shown in Figure 1b. This is in accord with the characteristics of intense a femtosecond laser pulse at its first propagation stage, as identified by Bergé et al. [10]. During the first stage, the beam intensity increases due to the optical Kerr effect focusing and the electron density being excited when the beam intensity becomes sufficiently large. Subsequently, the generated electron density has a threshold-like response, which limits the further increase of peak intensity inside the filament by defocusing the beam in turn [9]. When a balance of these two processes is achieved, the laser intensity inside the filament gradually stabilizes. This special phenomenon is usually named intensity clamping [9]. This light guide coupling with a significant electron density level is identified as the second stage by Bergé et al. [10]. Based on the values of the peak intensity, it can be clearly seen from Figure 1a that the clamping intensity in dry air is much higher than that in wet air. The clamping intensity of UV filaments decreased more obviously in wet air conditions compared with the results of

mid-infrared lasers filaments [15]. The peak intensity is maintained for several meters and is coupled with a significant electron density level (Figure 1b). The laser pulse tends to form longer filaments in wet air. Due to ionization of water vapor, the on-axis electron density of UV pulses in wet air is much greater than that in dry air, which can be clearly seen in Figure 1b. At a larger z, the pulse ultimately diffracts as ρ decreases to zero. In addition, the simulation results from the measured MPA parameters are very close to those from the theoretical calculation results in dry air (as shown with black dash lines).

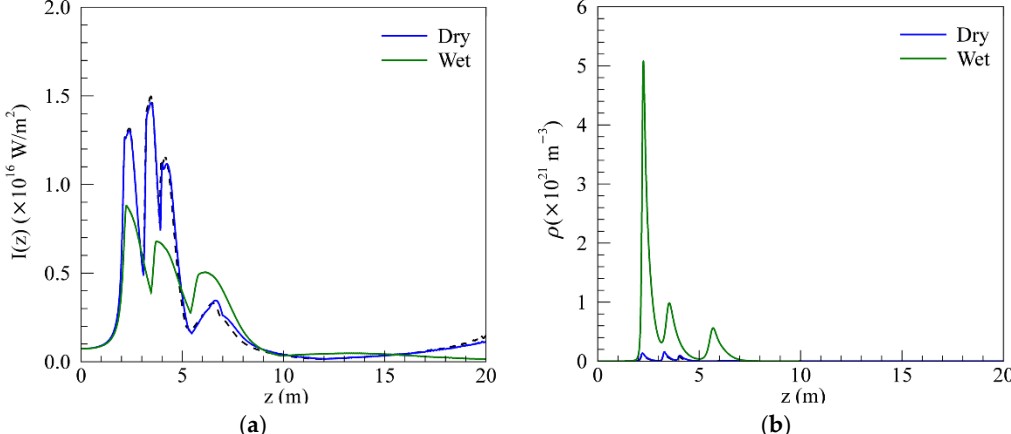

(**a**)　　　　　　　　　　　　　　　　　(**b**)

**Figure 1.** Evolution of the on-axis (**a**) intensity, (**b**) electron density as functions of z for a UV pulse propagating in air. The blue and green solid lines represent the simulated results in dry and wet air conditions, respectively. The cross section $\sigma^K$ for $N_2$, $O_2$, and $H_2O$ is obtained from the measured results by Shutov [17]. The dry air is assumed to only consist of $N_2$ and $O_2$ in a ratio of 0.78 to 0.21. For wet air, we impose a composition of 78.0% $N_2$, 21.0% $O_2$, and 1.0% $H_2O$ (gas). The black dash line represents the simulated results of dry air when its cross section $\sigma^K$ for $N_2$ and $O_2$ are obtained from the theoretical formula in Reference [9]. The input beam is a Gaussian shape with $P_{in}$ = 10$P_{cr}$, $w_0$ = 1 mm, and $\tau_p$ = 90 fs.

Figure 2 shows the number of electrons ionized from $N_2$, $O_2$, and $H_2O$. The number of electrons was calculated by $N_e(z) = \int_0^z dz \int_0^{r_{max}} dr \int_0^{2\pi} d\phi \rho(r, \tau_{max}, z) r \sin \phi$. The solid lines represent the results from the measured MPA parameters under wet conditions. It clearly shows that the number of electrons increased substantially in wet air compared with dry air, which demonstrates the remarkable contribution of $H_2O$ molecules to the air ionization. The MPA process is a crucial mechanism for the pulse's energy loss [14]. As shown in Figure 2b, the pulse energy decreases rapidly during the filamentation stage. The filamentation tends to deposit more energy in wet air conditions.

To investigate the underlying physics behind the filamentation for the two cases in the spatial regime, we plot the fluence distribution as a function of the propagation distance, which is shown in Figure 3. The fluence distribution is defined as $F(x, y, z) = \int_{-\infty}^{+\infty} |E(x, y, t)|^2 dt$. The several highlighted core areas in Figure 3 clearly indicate defocusing-refocusing cycles in the filaments [12]. There are mainly three cycles in dry air and wet air. The longer bright core in wet air further indicates that the laser pulse forms longer filaments in wet air. Compared with filamentation in dry air, the radius of filament gets slightly wider and the intensity of fluence becomes lower in wet air. These results are consistent with the experimental results reported by Shutov et al. [18].

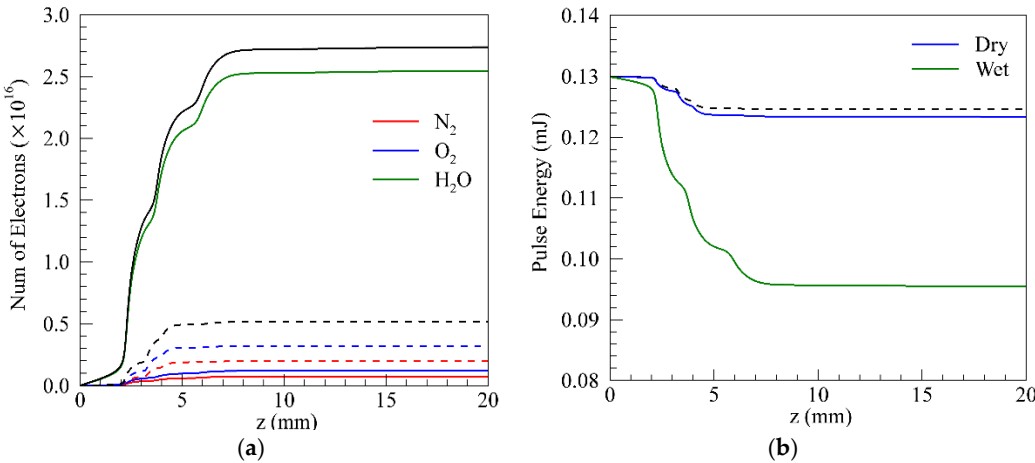

**Figure 2.** Evolution of (**a**) number of electrons ionized from $N_2$, $O_2$, and $H_2O$, and (**b**) pulse energy as functions of z. The number of electrons is calculated by $N_e(z) = \int_0^z dz \int_0^{r_{max}} dr \int_0^{2\pi} d\phi \rho(r, \tau_{max}, z) r \sin\phi$. In Figure 2a, the blue dash line is the results for $O_2$ gas, the red dash line denotes the results for $N_2$ gas, and the black dash line represents the results for dry air gas. The results are obtained from the calculated multiphoton absorption (MPA) parameters in Reference [9]. The solid lines in both figures are the results from the measured multiphoton absorption (MPA) parameters in Reference [17], which are clearly indicated in the legends, respectively.

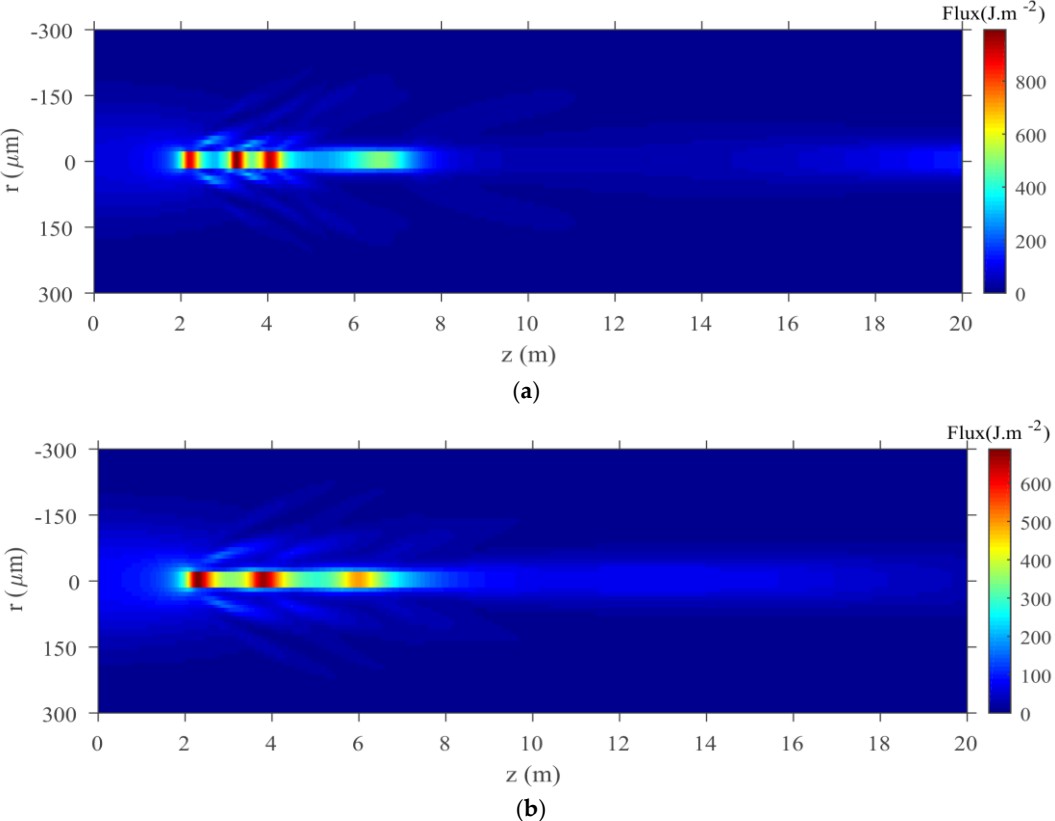

**Figure 3.** Evolution of the fluence distribution along the propagation distance. (**a**) Dry air and (**b**) wet air.

The evolution of the radius and intensity of the filament can also be clearly seen from the transverse energy flux distributions, as shown in Figure 4. The energy is concentrated into the central area with a slightly higher peak fluence in dry air (Figure 4a) than in wet air (Figure 4d) at the initial position of the filament. During the intensity clamping, an obviously higher peak fluence takes place in dry air

when propagation distance increases (Figure 4b,e). However, at the end of filament formation, the peak fluence in dry air becomes smaller than in wet air (Figure 4c,f). Relatively, the radius of filament is wider but its intensity is lower in wet air when compared with in dry air.

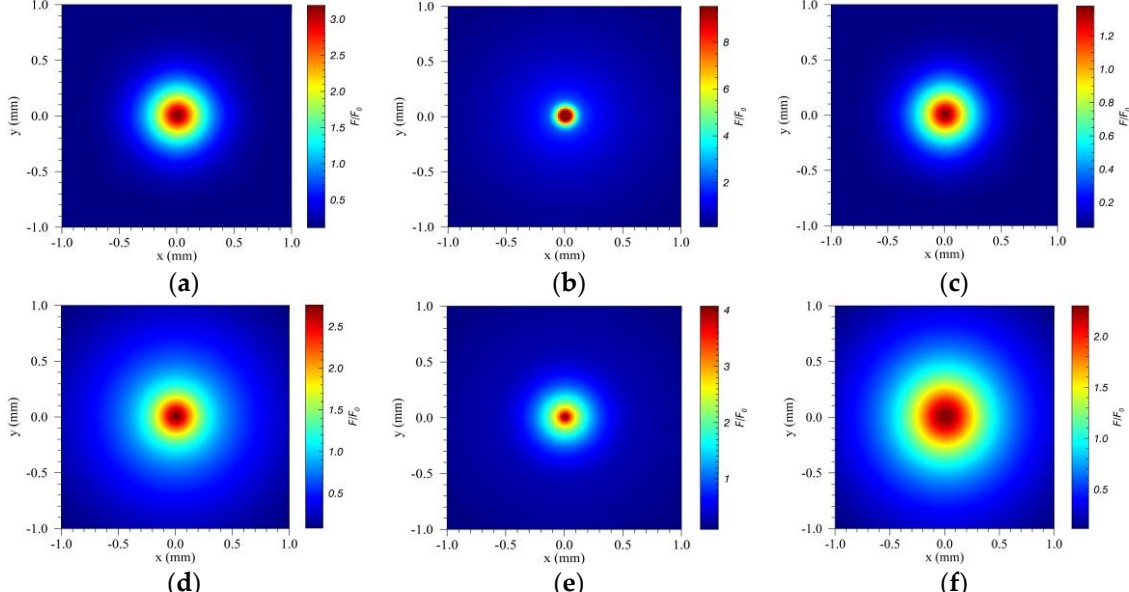

**Figure 4.** Fluence distribution $F/F_0(x,y)$ at different propagation distance z. The top row, (**a**) z = 2.0 m, (**b**) z = 3.5 m, and (**c**) z = 7.0 m for dry case. The bottom row, (**d**) z = 2.0 m, (**e**) z = 2.5 m, and (**f**) z = 7.0 m for wet case.

Figure 5a–d details the temporal distortions of the pulse at four different propagation distances. During the self-focusing stage, the pulse keeps its initial Gaussian shape and undergoes only an increase of its peak intensity (Figure 5a). The peak value in dry air is much higher than in wet air. Obvious multi-peaked structures and pulse broadening can be observed during the filamentation process, as noted in Figure 5b,c. The trailing pulse and leading peak in temporal profile is created by plasma defocusing [10,22,23]. The symmetric spikes of the pulse in time are associated with the GVD [24]. The GVD can further broaden the temporal width [14]. As shown in in Figure 5b,c, the number of splits in temporal profile changes little in both types of air gas. The degree of compression of a femtosecond pulse depends on air humidity. The broadening of laser pulses in dry air is much more obvious than in wet air. In the end of filamentation, the pulse propagated in wet air broadens and its peak intensity exceeds that in dry air (Figure 5d).

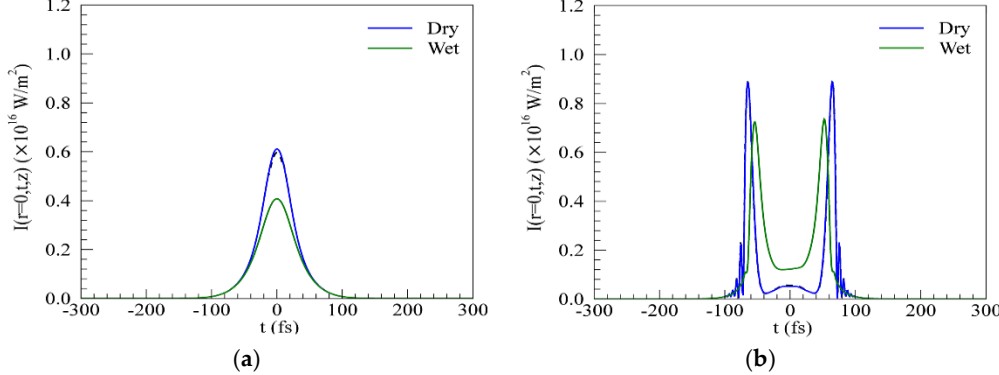

**Figure 5.** *Cont*.

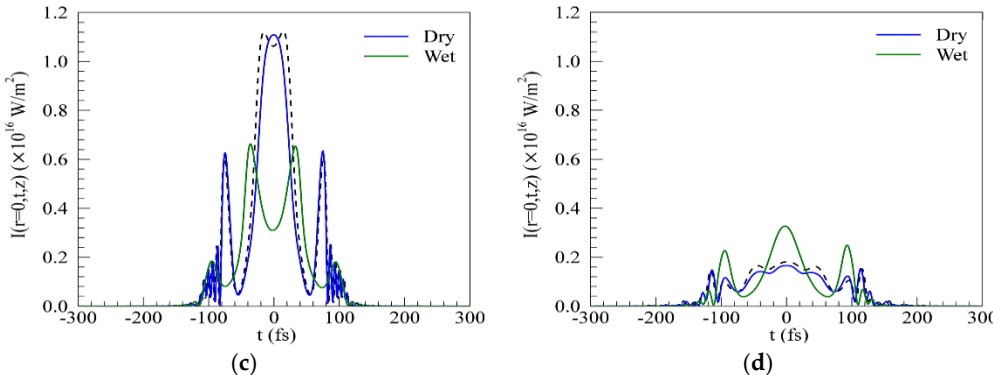

**Figure 5.** Temporal profile as functions of the propagation distance z. (**a**) z = 2.0 m, (**b**) z = 2.75 m, (**c**) z = 4 m, and (**d**) z = 5.5 m. The blue and green solid lines represent results from the measured MPA parameters in dry and wet air conditions, respectively. The dry air is assumed to only consist of $N_2$ and $O_2$ in a ratio of 0.78 to 0.21. For wet air, we impose a composition of 78.0% $N_2$, 21.0% $O_2$, and 1.0% $H_2O$ (gas). The black dash line represents the simulated result from the calculated MPA parameters in Reference [9].

In meteorology, relative humidity (RH) is an important physical measure to describe the amount of water vapor present in ambient conditions. We further investigated the nonlinear propagation of UV pulses in conditions with different relative humidity. The relative humidity grows as the amount of water vapor increases. By definition, relative humidity can be expressed as the ratio of vapor pressure (e) to saturation vapor pressure ($e_{sw}(T)$):

$$RH = \left[\frac{e}{e_{sw}(T)}\right]_{p,T} \tag{4}$$

The saturation vapor pressure of pure water is only a function of temperature, which can be calculated according to Wexler's formulations [25] as follows:

$$\begin{aligned}
e_{sw} = \quad & 0.01 \exp\Big[-2991.2729T^{-2} \\
& -6017.0128T^{-1} + 18.87643854 \\
& -0.028354721T + 0.17838301 \times 10^{-4}T^2 \\
& -0.84150417 \times 10^{-9}T^3 + 0.44412543 \times 10^{-12}T^4 \\
& +2.858487 \ln T\Big]
\end{aligned} \tag{5}$$

where $e_{sw}$ is in hPa and $e_0 = 6.11$ hPa is the saturated vapor pressure at 0 °C. $T$ is absolute temperature in K. For moist air, a slight correction must be made to calculate the saturation vapor pressure, which is expressed by:

$$e'_{sw}(p,T) = f(p) \times e_{sw}(T) \tag{6}$$

where $f(p)$ is called the enhancement factor and its formulation is given below [26]:

$$f(p) = 1.0016 - 0.074p^{-1} + 3.15 \times 10^{-6}p \tag{7}$$

Based on Equations (5)–(7), the saturation vapor pressure in moist air is 24.991 hPa at T = 21 °C. The vapor pressure e can then be derived from the ideal gas state equation:

$$pV = nRT \tag{8}$$

where $p$ is the pressure, $V$ is the volume, and $n$ is the amount of substance of the gas, respectively, which can be expressed as:

$$n = \frac{N}{N_A} \tag{9}$$

where $N$ is the number of gas molecules and $N_A$ is the Avogadro constant. As above, we find for cases with 10%, 75%, and 90% relative humidity, the relevant water content is 0.245%, 1.866%, and 2.248%, respectively. As for the case with 1.0% water content analyzed above, the relative humidity was 40%. In order to keep the total atmospheric density constant in all cases ($\rho_{N_2} + \rho_{O_2} + \rho_{H_2O} = \rho_{at}$), the components of dry air were reduced proportionally.

Figure 6 shows the on-axis intensity and electron density along propagation distance for the five different humidity values. In the simulation, we adopted $I_0 = 2.5 \times 10^{15}$ W/m$^2$, $\tau_p = 160$ fs, and r = 1 mm. We can clearly see that the atmospheric humidity levels do influence the overall filament dynamics. As the content of water vapor increases, the initial filamentation position slightly increases. The on-axis intensity experienced obviously decreases for 0% humidity to 40% humidity. The evolution of on-axis intensity in 75% humidity condition is very close to the 90% humidity condition. The differences in evolution of on-axis intensity along z decrease gradually from low humidity to high humidity. Analogously, the peak of on-axis electron density increases rapidly from low humidity to high humidity and the trends of increasing gradually decreases as the water content increases.

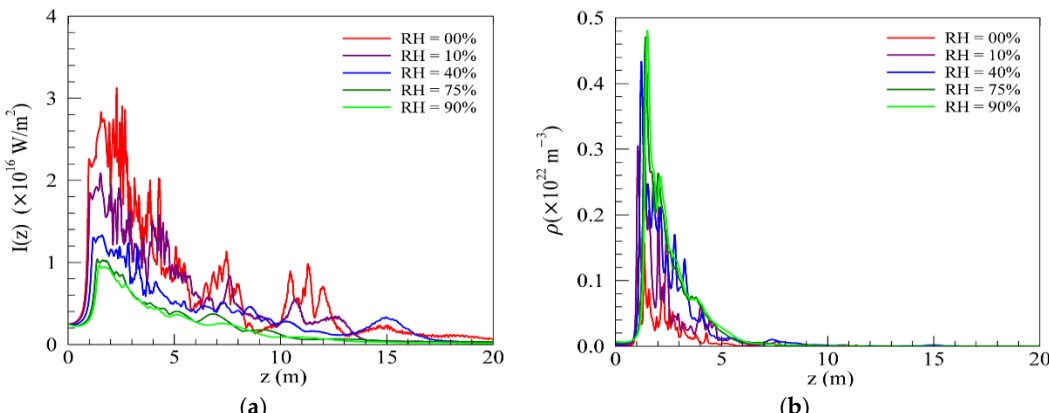

**Figure 6.** Evolution of the on-axis (**a**) intensity and (**b**) electron density along propagation distance z in different humidity conditions. Red solid lines: 0% humidity; purple solid lines, 10% humidity; blue solid lines, 40% humidity; green solid lines, 75% humidity; light green solid lines, 90% humidity.

Figure 7 shows the total number of electrons ionized from H$_2$O and the overall energy of the wave packet as a function of propagation distance z. As presented in Figure 7a, the number of electrons increases with the water content but the trend of increase gradually decreases as the humidity increases. In other words, the density of electrons tends to saturate in high humidity conditions. This saturation is attributed to the nonlinear energy loss in the multiphoton gas ionization by Shutov [17]. The differences in nonlinear energy loss can be clearly seen in Figure 7b. The deposited energy of filamentation increases with the humidity increases but the growth quantity takes on reducing trends in the higher humidity conditions.

We also simulated the femtosecond filamentation with initial peak intensity of $I_0 = 1.0 \times 10^{15}$ W/m$^2$ and $I_0 = 5.0 \times 10^{15}$ W/m$^2$. We find the conclusions are identical to those above and the results are not plotted again.

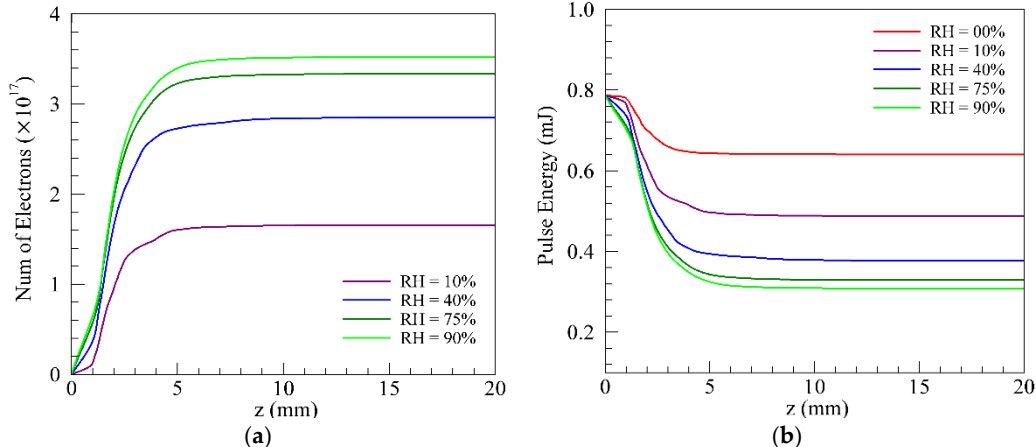

**Figure 7.** Evolution of (**a**) number of electrons ionized from $H_2O$ and (**b**) pulse energy along propagation distance. Purple solid lines, 10% humidity; blue solid lines, 40% humidity; green solid lines, 75% humidity; light green solid lines, 90% humidity.

## 4. Conclusions

In conclusion, we have presented a detailed numerical study of the UV filamentation processes in dry and humid air conditions. The results show that clamping intensity and electron density are very sensitive to the water vapor. Ionization of $H_2O$ molecules plays a significant role in the air ionization. The filamentation tends to deposit higher energy in wet air conditions. During the filamentation stage, the temporal profile is compressed narrower in wet air than in dry air. With the humidity increase, both the clamping intensity and electron density tend to increase. The number of electrons and pulse energy tend to be saturated in the higher humidity conditions. The above results clearly demonstrate the significant roles of photoionization of water vapor in research on UV filamentation.

**Author Contributions:** Conceptualization, L.L.; methodology, Q.Z.; software, K.Z.; investigation, C.W.; resources, X.X.; writing—original draft preparation, Q.Z.; writing—review and editing, S.H.; visualization, M.C.; supervision, T.G.

**Funding:** This work has been supported by the Fund of the National Natural Science Foundation of China (NSFC) (41575024, 41875025). These grants are greatly appreciated.

**Acknowledgments:** We acknowledge the help of Jingjing Ju from Shanghai Institute of Optics and fine Mechanics, Chinese Academy of Sciences.

**Conflicts of Interest:** The authors declare no conflicts of interest.

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
