# Peer review of "Numerical Investigation on the Influence of Water Vapor Ionization on the Dynamic and Energy Deposition of Femtosecond Ultraviolet Laser Filamentation in Air"

_applsci, doi:10.3390/app9204201_

Round 1

Reviewer 1 Report

The authors did calculation on 248nm driven laser filamentation in air with different water concentration. They show that clamping intensity and electron density are very sensitive to the water vapor due to the different total multiphoton absorption process. They also show that the temporal profile of the laser pulse is compressed narrower in wet air than in dry air. The paper is well written; however, some parts can be improved.

There is one principle question I have with your calculation. The ref. [2] measured the fluorescence spectra in a wet air filament at 800 nm and they can see a significant contribution of OH lines. How much is the absorption of 248 nm influenced by molecular dissociation of H20 ? Dissociation of H2O would also lead to a higher absorption. Is there an experiment like the experiment from Chin for 248 nm ? Is this comparable with your results

In line 117 the authors writhe that the water vapor has no influence on the self-focusing position. It would be better to tell the reader from where you get the conclusion. (maxima intensity of the intensity in fig 1.a). It would be nice to discuss the figure 1.a in respect to figure 1.b and then come up with this conclusion. (How you do it later in the paper.)

In 121 you wrote: Due to ionization of water vapor, the on-axis density of UV pulses … Its maybe better to write … the on-axis electron density …

Figure 3. Is it possible to increase the resolution of the calculation? It looks so that the flux brakes suddenly down at a certain r. Has this filament not a Gaussian form. (This looks strange to me) Please add a picture for a calculation for the xy plane as well.

Reviewer 2 Report

The Manuscript titled "Influences of water vapor ionization on the dynamic and energy deposition of femtosecond ultraviolet laser filamentation in air" describes numerically the influence of femtosecond laser propagation in the air with water vapor content in UV region. The numerical model is presented well with the results described correctly. However, this type of work seems to be important only when backup with experimental data. The authors talk about fs light in UV region which is very difficult to achieve. If the authors plan to do the experiments with fs UV light in future then I would appreciate the results be included. In addition, I find surprising so many email address in the affiliation with many email host not working. 

Reviewer 3 Report

The manuscript presents a numerical investigation on the influence of ionization of water vapor on nonlinear propagation of ultraviolet filaments. This is an interesting piece of work in which the formation and propagation dynamics of ultraviolet filaments in different humidity conditions are studied in detail.

Here are the main aspects that need to be corrected prior to publication:

Based on the fact that the manuscripts is a numerical investigation, this aspect should be clearly stated throughout the whole paper, starting with the Title itself. Therefore, please correct the Title of the paper to make clear to the potential readers, that the study they are about to read is a numerical study.

Please explain how the multiphoton absorption (MPA) curves shown in figure 1 were obtained. Solid lines are said to be obtained from measured parameters. Please give details about the experimental procedure performed to obtain such data.

Minor aspects:

Please correct letters in figure 4. Letters corresponding to the different plots should be a), b) c) and d).

In line 169: esw must be esw.

In line 188 (Figure 5 legend): the curve corresponding to 95% humidity should be called “light green solid lines”. Same happens in legend of Figure 6.

Round 2

Reviewer 1 Report

Most of the suggested corrections have been done. The resolution of fig. 3 is now better. However, I would suggest the authors to show one ore better two images in the xy direction for dry and wet air condition. One for a cut where the flux is low, and one where the maximum flux.   

Reviewer 2 Report

The manuscript "Numerical investigation on the influence of water vapor ionization on the dynamic and energy deposition of femtosecond ultraviolet laser filamentation in air" is modified well from the earlier version. The title clearly explains the numerical nature of the paper. However, it is still hard to imagine UV intensity of 10ˆ16W/cm2 at a wavelength of 248nm and 90 fs which is the assumption by the authors in their simulation. Therefore, I would encourage authors to at least mention the experimental plan or approach for such high intensity UV radiation which will help readers.  

Reviewer 3 Report

Authors answered to most queries posed by this reviewer.

There is still an open issue relative to the multiphoton absorption (MPA) curves shown in figure 1. Please include in the manuscript the explanation on how these curves were obtained. Solid lines are said to be obtained from measured parameters. Please give details in the manuscript about the experimental procedure performed to obtain such data.

Round 3

Reviewer 3 Report

Authors corrected the manuscript according to the queries posed by this reviewer. The manuscript is now acceptable for publication.